# A cryogen-free, semi-automated apparatus for bullet-dynamic nuclear polarization with improved resolution

Karel Kouřil[1], Michel Gramberg[1], Michael Jurkutat[1], Hana Kouřilová[1], and Benno Meier[1,2]

[1]Institute for Biological Interfaces 4, Karlsruhe Institute of Technology, Germany
[2]Institute of Organic Chemistry, Karlsruhe Institute of Technology, Germany

**Correspondence:** Karel Kouřil (karel.kouril@kit.edu) and Benno Meier (benno.meier@kit.edu)

**Abstract.** In dissolution-dynamic nuclear polarization, a hyperpolarized solid is dissolved with a jet of hot solvent. The solution is then transferred to a secondary magnet, where spectra can be recorded with improved sensitivity. In bullet-dynamic nuclear polarization this order is reversed. Pressurized gas is used to rapidly transfer the hyperpolarized solid to the secondary magnet, and the hyperpolarized solid is dissolved only upon arrival. A potential advantage of this approach is that it may avoid excessive dilution and the associated signal loss, in particular for small sample quantities. Previously, we have shown that liquid-state NMR spectra with polarization levels of up to 30 % may be recorded within less than 1 second after the departure of the hyperpolarized solid from the polarizing magnet. The resolution of the recorded spectra however was limited. The system consumed significant amounts of liquid helium and substantial manual work was required in between experiments to prepare for the next shot. Here, we present a new bullet-DNP system that addresses these limitations.

## 1 Introduction

Dissolution-dynamic nuclear polarization (D-DNP) can provide solutions of hyperpolarized molecules with near unity spin polarization, corresponding to signal enhancements of > 10,000 in state-of-the-art NMR instruments (Ardenkjær-Larsen et al., 2003; Ardenkjaer-Larsen et al., 2015; van Bentum et al., 2016; Ardenkjær-Larsen, 2016; Jannin et al., 2019; Kurzbach and Jannin, 2018; Zhang and Hilty, 2018; Ardenkjær-Larsen, 2019). Solutions containing hyperpolarized molecules such as pyruvate are now used in a number of clinical trials to measure human metabolism in vivo (Nelson et al., 2013; Comment, 2016; Wang et al., 2019).

For spectroscopic applications, the substantial polarization attainable with D-DNP does not in general translate into a corresponding sensitivity gain (Otikovs et al., 2019). This is because the analyte concentration is reduced in the dissolution step, and because the throughput of the dissolution-DNP experiment is typically much lower than the repetition rate that can be achieved when recording signals at or near thermal equilibrium. If for example a dissolution-DNP experiment yields a hyperpolarized spectrum with a 10,000 fold enhancement, but with a 100-fold dilution, it is possible to record a spectrum with the same signal-to-noise ratio by averaging 10,000 scans which is often feasible. In addition, the hyperpolarized spectra may display a lower resolution due to sample inhomogeneities that arise during the preparation. The radicals required for DNP may cause

fast relaxation in particular during the transfer process. The spectrum recorded without hyperpolarization will neither require a polarizer nor any staff operating it, and it may display better resolution and better reproducibility.

Several strategies have been developed to tackle the above mentioned issues. Radical-free solutions can be obtained using photo-induced radicals which are only stable at cryogenic temperatures - a D-DNP experiment performed with such polarizing agents naturally yields radical-free solutions upon dissolution (Capozzi et al., 2017; Eichhorn et al., 2013; Capozzi et al., 2021). The polarizing agent can also be fixed in a porous matrix which is then impregnated with a substrate. A solvent is used to flush the hyperpolarized substrate out of the matrix, leaving the radicals behind (Gajan et al., 2014; Cavaillès et al., 2018). Radicals such as TEMPOL may also be scavenged using ascorbate, and also glassing agents may be avoided by freezing solutions in isopentane (Lama et al., 2016). The rapid injection approach limits relaxation during sample transfer by reducing the time from dissolution to NMR acquisition (Bowen and Hilty, 2010; Katsikis et al., 2015). In the dual-solvent approach one can select such a radical that it preferably dissolves in the second solvent (Harris et al., 2011). This approach also addresses the issue of sample dilution, however at a cost of reduced resolution. Another approach to reduce the dilution of the analyte in the final solution is to polarize more material to begin with. This strategy is costly for precious samples, but viable in particular for experiments involving hyperpolarized water (Olsen et al., 2016; Szekely et al., 2018; Novakovic et al., 2020).

The rapid melt DNP (Sharma et al., 2015) avoids the dissolution altogether and is also readily combined with NMR on a chip (van Meerten et al., 2018; de Vries et al., 2019). The throughput of a DNP polarizer can be increased by parallel polarization of multiple samples (Batel et al., 2012; Cheng et al., 2020), or by employing [1]H DNP in combination with cross polarization (Jannin et al., 2012) or via adiabatic demagnetization (Jacquinot et al., 1974; Elliott et al., 2020).

Despite its limitations D-DNP has found applications in metabolomics (Dey et al., 2020), diffusion ordered NMR spectroscopy (Guduff et al., 2017), and may be used for the extraction of carbon-carbon couplings at natural abundance (Otikovs et al., 2019). In special cases, long-lived states may be used to extend the time over which hyperpolarized spin order can be used (Bornet et al., 2014; Jhajharia et al., 2017; Levitt, 2012; Elliott et al., 2018). On the other hand, it is possible to rapidly record multi-dimensional spectra with fast NMR techniques (Giraudeau et al., 2009; Schulze-Sünninghausen et al., 2014; Schanda and Brutscher, 2005).

We have recently presented a new approach to dissolution-DNP, named bullet-DNP (Kouřil et al., 2019), in which the hyperpolarized solid is ejected rapidly from the polarizer using pressurized gas, and dissolved only upon arrival in the target magnet. A key advantage of this approach is that it is scalable towards small solvent amounts. We have reported a less than 10-fold dilution for a hyperpolarized substrate volume of 80 μL, and we have also reported experiments with substrate volumes as small as 10 μL (Kouřil et al., 2018). Since typical D-DNP systems lead to a 100-fold dilution for small sample quantities, bullet-DNP achieves - other things being equal - a 10-fold increase in signal intensity. At this point, one would have to average 1,000,000 scans at thermal equilibrium, which is not practical anymore.

However, the spectra reported in our previous work displayed a linewidth of > 30 Hz due to gas bubbles that were incorporated into the solution upon the impact of the bullet on the solvent. The low resolution has rightly been pointed out as a significant limitation of bullet-DNP (Pinon et al., 2020), since it greatly reduces the sensitivity and often prohibits the extrac-

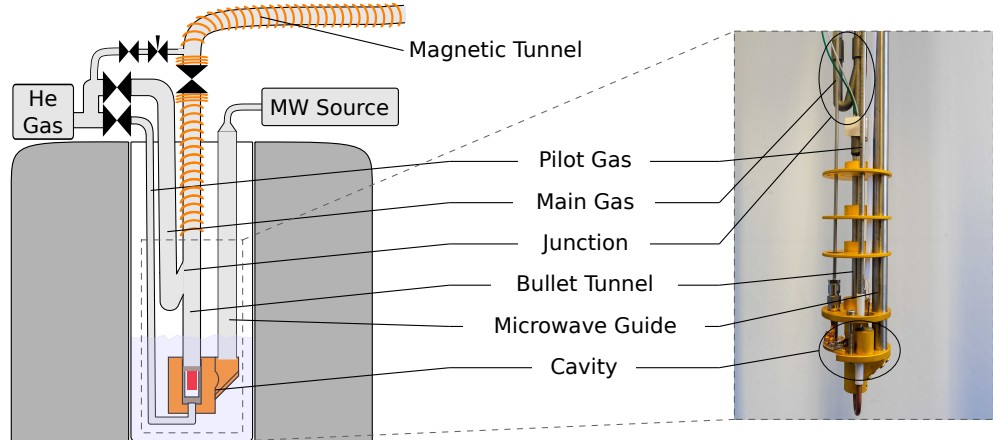

**Figure 1.** Sketch (left) and photograph (right) of the DNP insert. The figure has been adapted from (Kouřil et al., 2019). In contrast to the previous design, the new insert uses not one, but two lines to supply the helium gas used to propel the sample: A 1/8" line (Pilot Gas) is used to lift the sample out of the isothermal zone of the cryostat. A substantially larger flow is then applied via a 1/4" line (Main Gas). This line is connected to the Bullet Tunnel via a 3D printed junction, in such a way that the helium flow is directed upwards as it enters the Bullet Tunnel.

tion of structural or compositional information from the obtained spectra. Other key limitations of our previous design were its substantial consumption of liquid helium (up to 100 L / week), and a labour intensive cleaning process between experiments.

In this bulletin we present a new system that we designed at our new home, Karlsruhe Institute of Technology, in order to address these limitations. Using a back pressure technique described first by Bowen et al. (Bowen and Hilty, 2010; Katsikis et al., 2015), we observe a $^{13}$C linewidth of down to $\sim 2$ Hz, using only 500 µL of aqueous solvent. Compared to our previous work, the amount of manual work required for cleaning the injection device in between shots is reduced substantially. The DNP magnet system is very similar to the one described first by Ardenkjær-Larsen and co-workers (Ardenkjær-Larsen et al.,

2018). A key advantage of this system is that it does not consume liquid helium. This magnet system is available commercially from Cryogenic Limited, UK, and has been installed successfully in several labs (Baudin et al., 2018; Kress et al., 2021).

## 2   Experimental

The bullet-DNP system comprises a cryogen-free magnet hosting the DNP insert (Fig. 1), an injection device inserted into the narrow bore magnet of a 400 MHz Bruker NMR spectrometer (Fig. 2), and a magnetic tunnel connecting the DNP insert to the

injection device.

### 2.1   Polarizer and DNP Insert

The magnet system (Cryogenic Ltd., UK) uses a closed helium cycle to cool the superconducting magnet and the sample space. A single cold head provides cooling power for the magnet and the sample space. The cold head is connected to an external

reservoir (200 L) from which gaseous helium can be condensed. Following condensation, the liquid helium may be transferred into the sample space via a needle valve. The sample space is connected to a dry Roots pump and achieves a base temperature of approximately 1.4 K. The outlet of the pump is connected to the external reservoir, thus closing the cycle. The magnet itself has a bore size of 50 mm. The maximum field strength is 9.4 Tesla, but the magnet is operated at 6.72 Tesla, corresponding to a microwave frequency of 188 GHz. The microwave source (Virginia Diodes Inc) has a nominal output power of 200 mW. In the experiments described in this paper, we operate the source at a fixed frequency, with a (non-optimized) output power of approximately 150 mW.

In our previous design, the sample was ejected using a burst of ambient temperature helium gas that was connected to the bottom of the DNP insert via a 1/4" tube. This burst of helium led to a substantial boil-off of liquid helium. Since the new cryogen-free system can only condense approximately 0.1 L of liquid helium / hour, we have redesigned the DNP insert to minimize the boil-off during the sample ejection. In the new design, shown in Fig. 1, only a small stream of helium is directed to the bottom of the DNP insert via approximately 1 m of 1/8" tubing. A second stream is connected to the transfer tube above the liquid helium level via 1/4" tubing, and provides the bulk of the helium gas used to propel the sample once it has left the bottom of the DNP insert. Both lines are controlled with air-actuated Swagelok diaphragm valves. To eject the sample, the valve on the 1/8" line is opened first, and the valve on the 1/4" line is opened with a delay of 50 ms. In order to avoid air ingression during sample insertion and ejection, we apply helium flush gas above the ball valve at the top of the DNP insert whenever the latter is open. This flush gas represents an additional heat load which currently dominates the heat load during the sample ejection. Based on the increase of helium pressure in the external reservoir, we estimate that approximately 10 mL of liquid helium are boiled off during both the sample insertion and ejection.

## 2.2 Tunnel

In order to preserve the nuclear spin polarization, we again use a solenoid that is wound around the entire transfer tube. For the sample transfer, this solenoid is energized with a current of 50 A, providing a field of approximately 60 mT. The solenoid starts and ends where the magnetic fields of the polarizer and the 400 MHz magnets drop below 100 mT, and extends well into either magnet. While permanent magnets provide substantially larger fields (Milani et al., 2015), we currently continue to use solenoids only, since they are readily adjusted to changes in the apparatus design.

The polarity of the magnetic field inside the polarizer can be set freely, and we use an anti-parallel alignment of the magnetic fields in the polarizer magnet and the secondary magnet. With this arrangement the orientation of the magnetic field with respect to the hyperpolarized solid may be preserved during the transfer, and there is no need for the transverse coils used in our previous setup (Kouřil et al., 2019).

## 2.3 Injection Device

The new injection device, shown in Fig. 2 arguably represents the most significant improvement over our previous work. The device has three central components: a solvent reservoir, a pinch valve and a flow NMR tube. The hyperpolarized solid is dissolved in the solvent reservoir, and the pinch valve controls the flow of liquid from the solvent reservoir to the NMR tube.

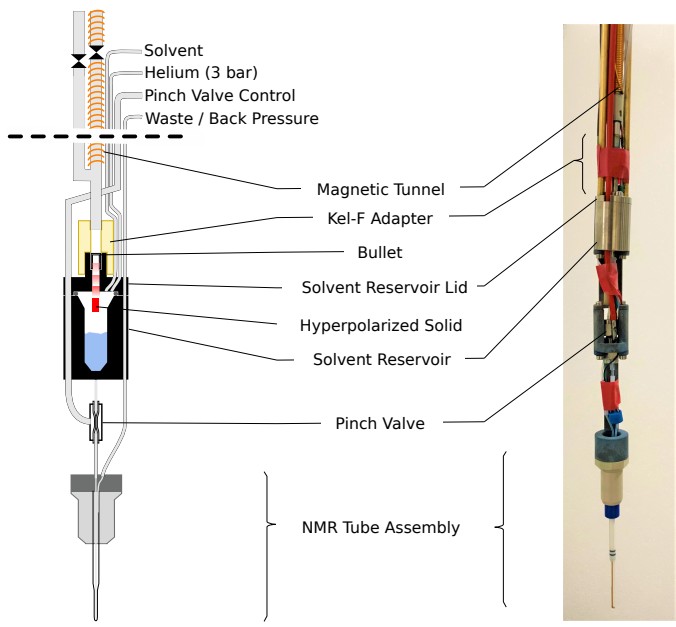

**Figure 2.** Sketch (left) and photograph (right) of the injection device. Prior to ejection the solenoid on the injection device sample tube is energized. The ball valve (shown between two sections of magnetic tunnel) is opened and the bullet is ejected from the DNP insert. The bullet arrives via a central steel tube, and comes to a stop at a constriction in the solvent reservoir lid. The hyperpolarized solid (red) passes through this constriction and dissolves in the solvent (blue) upon impact. The solvent reservoir is pressurized, and the pinch valve is opened, causing the liquid to flow into the NMR tube assembly via its inner capillary. After another delay, back pressure is applied to the exit line of the NMR tube assembly and NMR acquisition is started. The empty bullet may be ejected after the experiment by pressurizing the solvent reservoir.

The bullet containing the hyperpolarized solid arrives through the magnetic tunnel and stops inside a constriction in the lid of the solvent reservoir. The hyperpolarized solid itself passes through the constriction and dissolves upon impact with the solvent in the reservoir. After the arrival of the bullet in the injection device, the ball valve at its top is closed, and the solvent reservoir
is pressurized via the tee in the main tube and a 1/16" tube connected directly to the reservoir, using helium at 3 bar. In this way the reservoir is rapidly pressurized also if the empty bullet is deformed during the impact in such a way that it blocks the channel in the solvent reservoir lid (see Fig. D1 for a photograph of the bullets before and after the shot). After a delay for settling, the pinch valve is opened briefly to fill the NMR tube. Back pressure at the outlet of the NMR tube is applied, and the NMR acquisition is triggered.
After the experiment, the magnetic tunnel is manually disconnected from the top of the injection device, and a disposal program applies positive pressure to the solvent reservoir, leading to the ejection of the empty bullet. At this stage, the magnetic tunnel is reconnnected. A cleaning program automatically cleans and dries the device by pushing the liquid out of the NMR tube assembly using pulses of positive helium pressure on the tube inlet. The helium gas is switched on and off for 6 s each,

and this cycle is repeated three times. Following this procedure, the system appears to be clean and dry, ready for the next experiment.

The 5 mm standard NMR tube that was used in the previous injection device has been replaced with a flow tube. The flow tube, developed jointly by Guy Lloyd-Jones and TgK Scientific Ltd, is available commercially from the latter. It has an outer diameter of 3 mm, and a small inner capillary that extends to the bottom of the tube. While the smaller diameter reduces the attainable sample volume inside 5 mm NMR probe heads, it ensures that the sample can simply be pushed out of the NMR tube by applying positive pressure through the inner capillary. The inner capillary of the tube assembly is connected to the solvent reservoir via the pinch valve.

The pinch valve consists of a short section of 1/8" silicone tube inside a sealed enclosure. Applying pressure in the enclosure collapses the silicone tube, closing the valve. The liquid flows into and out of the valve via 1/16" Tefzel tubes, which are pushed into the silicone tube such that 2 cm at the center of the silicone tube are free to collapse. This part of the silicone tubing runs inside the enclosure which is made from a 1/8" Swagelok union. The ferrules of the union squeeze both the silicone and the Tefzel tubes, providing a good seal also with positive pressure in the 1/16" tubes. The enclosure is pressurized with air via a small hole in the side of the Swagelok union, that is brazed to a short piece of 1/8" steel tube. This steel tube is pushed into a 4 mm OD plastic tube which runs to the top of the injection device. A pressure of 8 bar is used to close the pinch valve.

The solvent reservoir is made from two titanium parts, similar to a bucket with a lid. An o-ring is used between the bucket and the lid. The reservoir has a capacity of approximately 2 mL, with a Luer taper at the bottom. The reservoir can be equipped with a heater and a temperature sensor. Three 1/16" Tefzel lines connect to the solvent reservoir via the lid. One line is used for loading the reservoir with solvent, and one line is used for pressurizing it. The third line can be used to supply a different solvent, e.g., for cleaning or activating. This line has not been used in the experiments reported here, and it is not shown in Fig. 2. A constriction in the solvent reservoir lid retains the bullet upon arrival, but allows the hyperpolarized solid to travel further until it hits the solvent inside the reservoir. A small adaptor piece made from Kel-F connects the solvent reservoir lid to the main steel tube that connects to the top of the injection device. To detect the arrival of the bullet, an LED and the photo-diode of a commercial photo interrupter (OPB350W250Z) are repackaged into a 3D printed holder that slides on the Kel-F adaptor. The photo-current through the photo-diode is used to detect the arrival of the bullet. A 2 mm hole is drilled into the side of the main steel tube near its bottom end, and connected to a 1/4" nylon hose via a tightly fitting PEEK piece. This vent tube ensures that the bullet can arrive at its destination at sufficient speed. At the injection device top, a Swagelok tee facilitates pressurizing the vent tube and the tunnel above the solvent reservoir. A ball valve above the tee is used to close the top of the main tube before it is pressurized. Lastly, another Swagelok union on top of this ball valve has six 1.0 mm holes drilled into its sides. Without this additional vent, the pressure wave from the impact of the bullet on the solvent is too strong for the pinch valve, causing an uncontrolled leakage of solvent into the NMR tube.

The injection device is loaded with solvent using a Cadent3 syringe pump that is connected to the control PC via a serial-to-ethernet bridge (Moxa).

## 2.4   Control

All valves are controlled via SMC pilot vales, which in turn are controlled via MIC2981 source drivers switched by an Arduino Mega. Pulse programs are stored using JavaScript Object Notation (JSON) and transferred and processed on the Arduino using the Arduino JSON library. On the control PC, a PyQT5 based graphical user interface facilitates control of the entire system.

## 2.5   Bullets

The bullets used in this experiment are consumables, a new bullet is used for every experiment. The bullets are made from PTFE on a Haas ST10 lathe. The lathe is equipped with a bar puller and parts catcher, such that it can machine approximately 70 bullets in an hour without human intervention. While this degree of automation may seem excessive, the high reproducibility and the tight tolerances afforded by the automated production process have greatly benefitted the debugging of the injection device, for which we consumed several hundred bullets. We have tried a range of other plastics for the bullets, such as PEEK and Vespel, but these tend to become too brittle at cryogenic temperatures.

## 2.6   Operation

For the actual DNP experiment a solution of of 15 mM OX063 in $1-^{13}$C-labeled pyruvic acid is prepared. A bullet is filled with 50 μL of this solution and placed carefully into a dewar with liquid nitrogen, such that the liquid nitrogen does not mix with the material during the freezing process. Subsequently, the frozen bullet is inserted into the polarizer against a small stream of helium gas, using a small funnel attached to a piece of 1/4" tubing. The funnel is disconnected, and the magnetic tunnel is connected manually. Once the sample is polarized, the control software on the PC is used to close the pinch valve on the injection device and fill the solvent reservoir with 500 μL of H2O/D2O 9:1, using a syringe pump. A second routine on the controller energizes the magnetic tunnel, transfers the program for bullet ejection and NMR tube loading to the Arduino, and triggers its execution.

First a stream of helium gas is flown through the magnetic tunnel to prevent air ingress into the DNP insert. Next the ball valve on the DNP insert is briefly opened and the bullet is ejected by applying the two streams of helium gas. After the arrival of the bullet at the injection device the ball valve at its top is closed and the solvent reservoir is pressurized with helium to 3 bar. After 1 s delay for settling the pinch valve is opened for 100 ms to fill the NMR tube. Back pressure (also 3 bar helium) is applied to the outlet of the NMR tube and a trigger signal is sent to the NMR spectrometer. The NMR aquisition starts 1.7 s after the ejection of the bullet from the polarizer. A $^{13}$C FID is recorded with $^{1}$H-decoupling every 2 s. 20 s after the NMR trigger the back pressure is released, but the NMR acquisition continues. After the experiment the empty bullet is ejected from the injection device and the NMR tube is cleaned as described above.

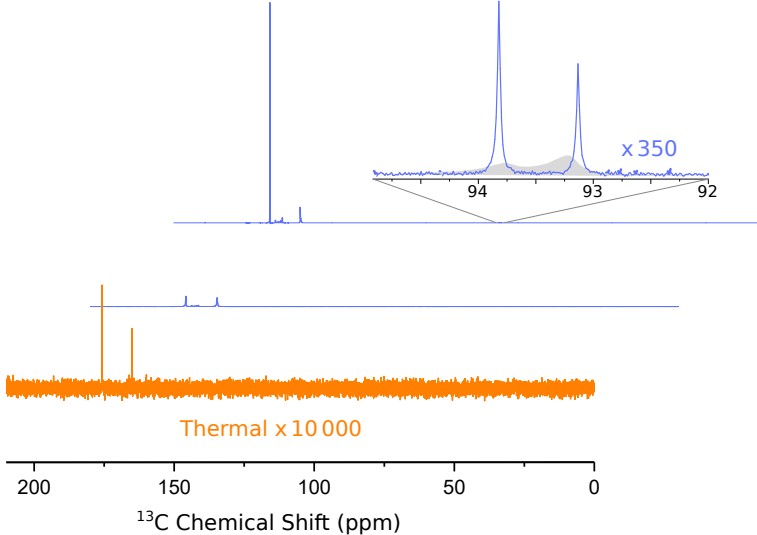

**Figure 3.** Hyperpolarized $^{13}$C spectra (blue lines) of 1-$^{13}$C pyruvic acid. The middle and upper line show the first two spectra, recorded 1.75 s and 3.75 s after the ejection of the bullet from the polarizer, respectively. A detailed assignment of the peaks has been given in our previous publication (Kouřil et al., 2019). The inset shows a magnified view of the doublet near 93 ppm, which is due to the natural abundance carbonyl of pyruvate hydrate. The asymmetry of this doublet yields a $^{13}$C polarization of - 22 % at the (labeled) carboxyl carbon. The full-width at half maximum of both peaks of the doublet is 3.2 Hz. The filled grey area in the inset shows the spectra from our previous publication for comparison (where we used positive DNP leading to an inverted asymmetry). The improvement in resolution is obvious. Note that in our experiment the line near 94 ppm is more intense since we used negative polarization.

## 3   Results

In Fig. 3 we show the first two $^{13}$C NMR spectra obtained in this experiment. The spectra were recorded every 2 seconds with $^1$H decoupling and a flip angle of 10 degree. The acquisition time was 1.75 s, and no line-broadening was applied during processing.

An analysis of the doublet near 93 ppm in the second spectrum yields a $^{13}$C polarization level of 22 %. Also shown in Fig. 3 is our previously published spectrum (gray envelope). The new system yields a full width at half maximum (FWHM) of 3.2 Hz in the second spectrum, corresponding to an order of magnitude increase in resolution compared to our previous work. The FWHM in the first spectrum is approximately 14 Hz. After the second spectrum the resolution keeps improving slightly as the solution settles in the NMR tube. The FWHM is below 2 Hz 14 s after the start of the NMR acquisition. The resolution does not change significantly when the back pressure is released 20 s after the beginning of data acquisition. The FWHM values for the first 11 spectra are shown in Fig. A1.

The limited resolution in the first spectrum is a reproducible effect which we attribute to a single large gas bubble that rises inside the NMR tube after injection of the solution. This gas bubble may be avoided if the NMR tube is evacuated prior to sample injection. Preliminary data obtained with this scheme in two consecutive experiments are shown in Fig. B1. The

injection device was not removed from the magnet between the two experiments. In the second experiment, we observe a linewidth of 4 Hz in the first scan, and approximately 2 Hz in the following scans. Based on the asymmetry of the doublet, the $^{13}$C polarization level in the second experiment is $27\pm1$ %.

## 4  Discussion

We believe that the new system described in this paper represents a substantial improvement over our previous work.

We estimate that the dual drive system boils off approximately 20 mL of liquid helium for one cycle of bullet loading and ejection. We believe that the heat load on the system is dominated by ambient temperature flush gas that is applied whenever the ball valve on the DNP insert is opened. Currently this valve is opened for several seconds during both sample loading and ejection. This heat load imposes a limit on throughput of the system. From the pressure changes in the external reservoir during helium condensation we estimate that the cryocooler can condense approximately 100 mL of liquid helium per hour which would limit the system to 5 shots per hour. For a high degree of carbon polarization we polarize for 1 - 2 hours, so that the overall rate is currently limited by the DNP step. However, the cooling power of the system will become relevant when several bullets are polarized at the same time, or when the polarization step is accelerated using cross-polarization.

The improvement in resolution is most likely due to the reduction of the amount of bubbles in the NMR tube. During the injection a single bubble forms near the bottom of the NMR tube, and flows to the top of the tube within 1 to 2 seconds. The amount of small bubbles or foam is substantially reduced when compared to the previous setup and the injected liquid appears clear immediately after the injection, indicating the absence of fine bubbles. Interestingly the resolution of the spectra does not deteriorate significantly when the back pressure is released 20 s after the sample injection. This suggests that, at this point, the bubbles no longer broaden the NMR lines.

In the experiments reported here, the solution is allowed to settle in the solvent reservoir for over 1 second from the bullet impact to the opening of the pinch valve. Moreover the bubble which forms during the injection degrades the quality of the first recorded FID. The two factors combined mean that the time from the sample ejection from the DNP magnet to the start of the acquisition of the first high-resolution spectrum is 3.75 seconds. The formation of an air bubble may be avoided by evacuating the NMR tube prior to injection of the liquid. Preliminary data show that resolutions of 4 Hz and 2 Hz are achievable 2 s and 4 s after sample ejection, respectively.

The injection device is able to perform multiple experiments without removal from the NMR magnet. After a shot, the empty bullet is ejected by pressurized gas and the fluid path is washed and dried automatically. While in principle the setup can run indefinitely, in practice the bullets do get damaged during the ejection. As a result, small fragments of PTFE can fall into the solvent reservoir and accumulate there. Over time this increases the probability that one or several of the fragments get sucked into the tubing and block the fluid path. Additionally, in approximately 1 in 10 shots the bullet breaks into larger fragments which then cannot be removed by the pressurised gas and remain on top of the solvent reservoir. In such cases, the injection device needs to be removed from the NMR magnet, cleaned manually and inserted back. The system can therefore currently

not run unattended experiments around the clock. However, the current level of automation is already rather convenient and constitutes a significant step towards a fully automated system.

At this point, the reproducibility of the spectral resolution is not very high. A linewidth of 5-6 Hz can be achieved routinely if the injection functions correctly. If the fluid path is blocked due to PTFE fragments, the resolution deteriorates quickly, and the injection device has to be removed from the magnet for cleaning. We are currently adjusting the bullet path to further reduce fragmentation. We have also observed radiation-damping, which likewise reduces spectral quality.

## 5   Conclusions

In conclusion, we have shown that bullet-DNP is compatible with cryogen-free magnets, and that hyperpolarized $^{13}$C spectra can be recorded on aqueous solutions with a resolution of 2 ppb (or 2 Hz at 9.4 Tesla) at a polarization level of 27 %. The system is highly automated, but cannot yet run subsequent experiments without human intervention. We believe that the system is now suitable for a range of spectroscopic applications, and hope to show first examples in the near future.

*Data availability.*

The NMR data shown in Fig. 3 are available for download in TopSpin format from doi:10.5445/IR/1000135039

## Appendix A:  Spectral Resolution versus Time

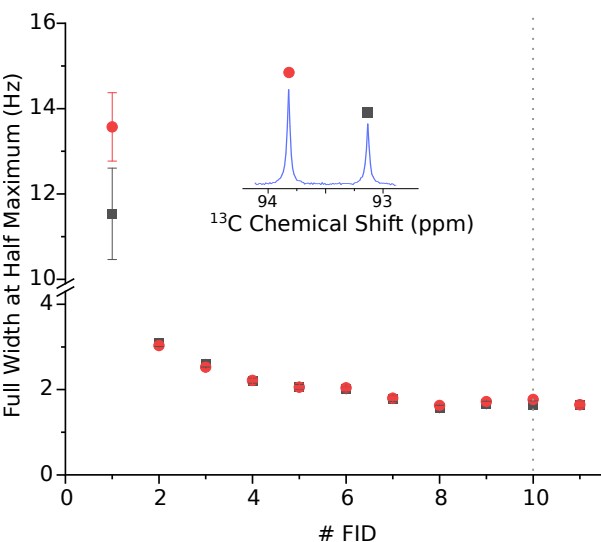

**Figure A1.** Full width at half maximum of the $^{13}$C doublet due to the carbonyl $^{13}$C atom of pyruvate hydrate for the data shown in Fig. 3. The linewidth of both peaks of > 10 Hz in the first scan is attributed to a gas bubble that rises up the NMR tube during the first scan. The second spectrum, recorded 3.75 s after the sample ejection displays a substantially improved resolution of 3.2 Hz. Over time the resolution improves further, and is better than 2 Hz 14 s after the beginning of the NMR acquisition. Note that there is no significant change in resolution when the back pressure is released 20 s after the beginning of data acquisition (i.e. near the beginning of FID # 10, marked with the dotted line).

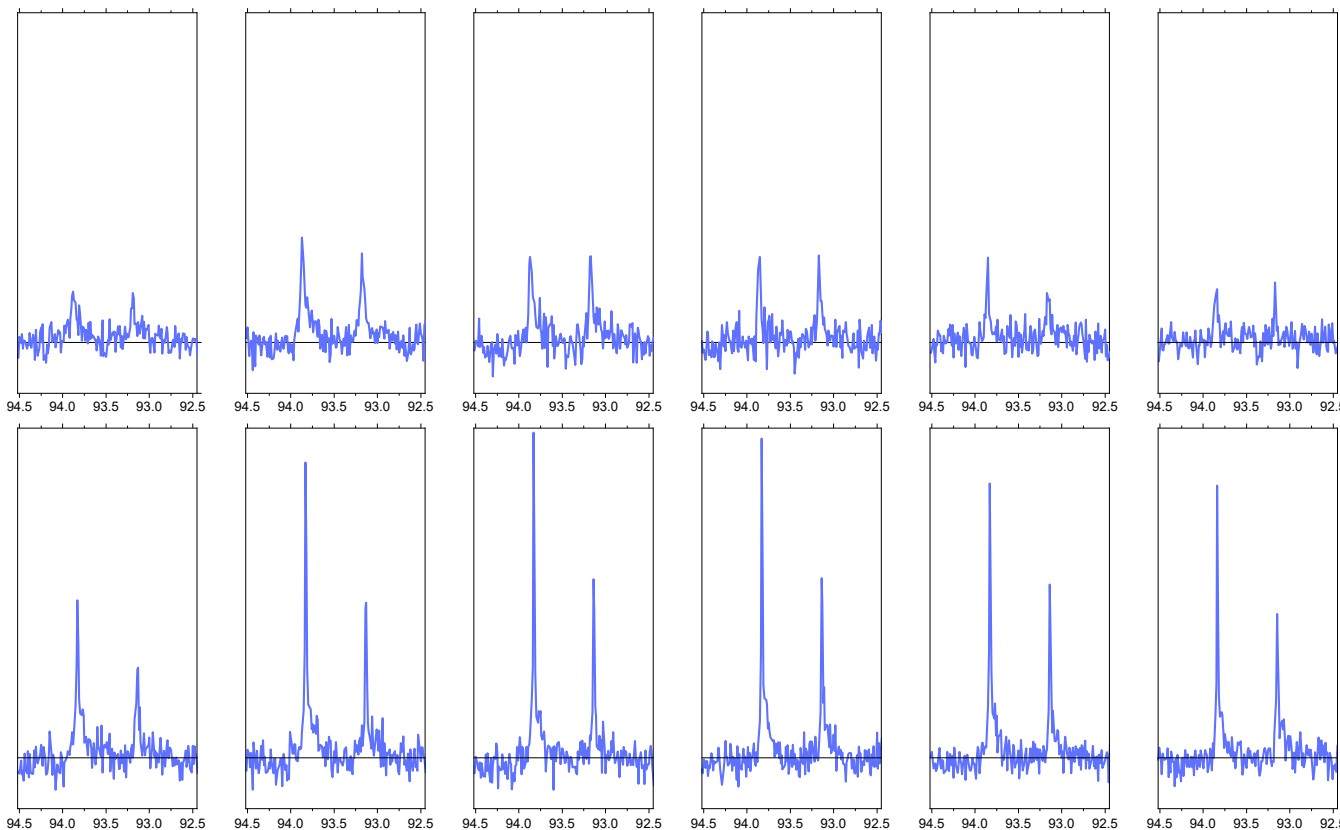

**Figure B1.** The first six NMR spectra of hyperpolarized 1-$^{13}$C pyruvate in aqueuous solution, recorded during two consecutive experiments (top and bottom row). Compared to the experiment shown in Fig. 3, the spectra were recorded with a reduced receiver gain. Only the spectral region corresponding to the carbonyl $^{13}$C of pyruvate hydrate is shown. For the experiments shown here, the sample tube is evacuated prior to solvent injection. Furthermore, the delay for settling (i.e. prior to the column labelled NMR in Fig. C1) is increased from 100 to 350 ms, so that the total time from sample ejection to the beginning of acquisition is 2 s. In the first experiment (top row), the polarization is low because the microwave irradiation frequency was not set correctly. In the second experiment, the microwave frequency was corrected. For the second dataset, the linewidth (FWHM) is 3.9(2) Hz in the first spectrum, 2.4(0) Hz in the second and 2.1$\pm$0.3 Hz in the subsequent spectra. The injection device was not removed from the magnet between the two experiments.

## Appendix C: Pulse Program for Sample Ejection, Dissolution and Injection

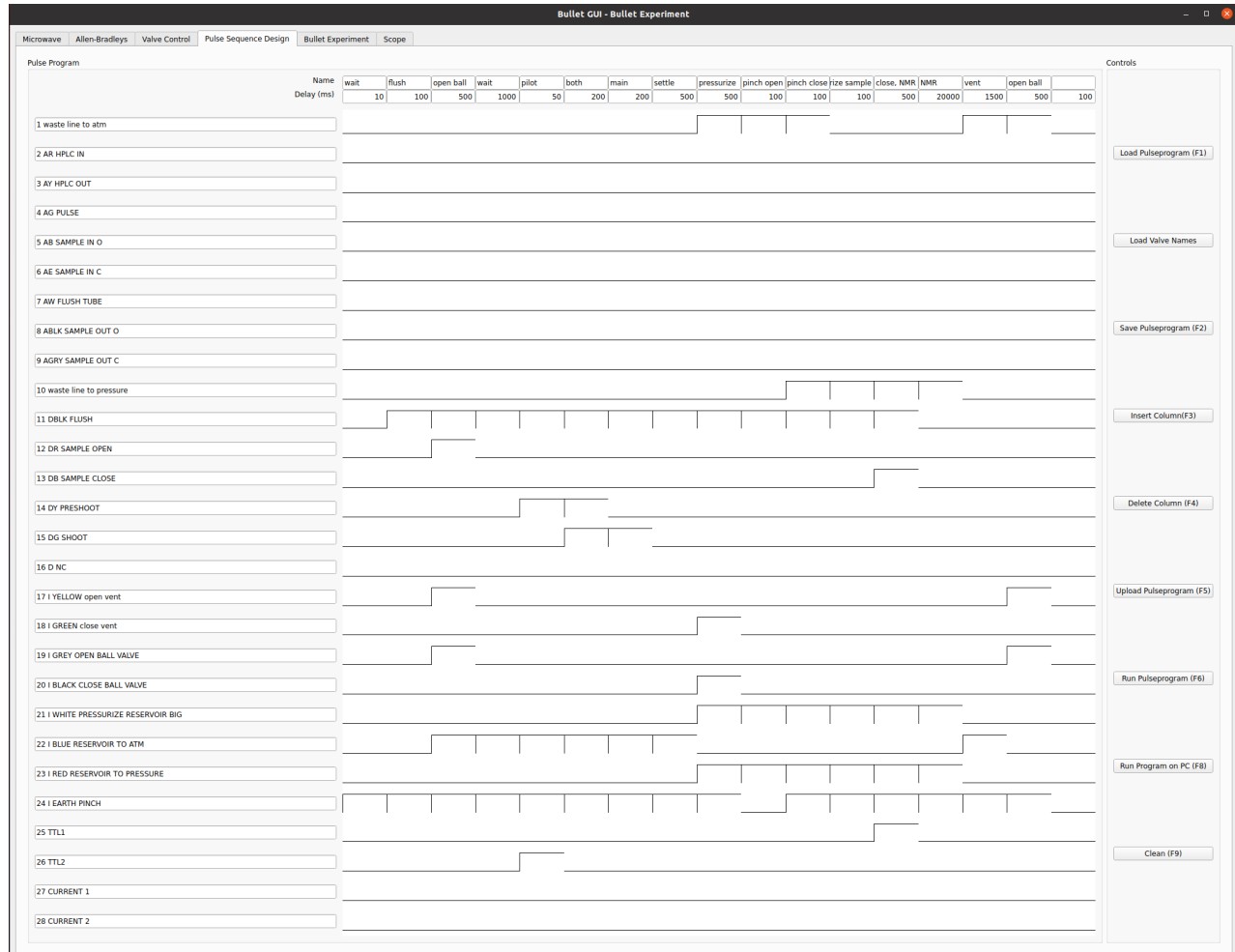

**Figure C1.** Bullet pulse program for sample ejection, dissolution and injection. The lines 1 to 24 correspond to digital output lines that are connected to pilot valves. The program runs through a sequence of 17 steps. The duration of each step is given in the top row, under its name. Valve 1 connects the outlet of the NMR tube to the atmosphere. Valves 2-9 are not used. Valve 10 connects the output line of the NMR tube to back pressure. Valve 11 applies flush gas on top of the DNP insert, and valves 12 and 13 are used to open and close the ball valve on the DNP insert top, respectively. Valves 14 and 15 apply helium drive gas to the DNP insert via the pilot and main line, respectively, and are used to propel the sample. Valves 17 and 18 open and close a ball valve that is connected to the vent line of the DNP injection device. Valves 19 and 20 open and close the ball valve on the injector. The line above the solvent reservoir can be pressurized with helium (3 bar) using valve 21. Helium pressure can also be applied directly to the solvent reservoir via valve 23. Finally valve 24 controls the pinch valve, which is only opened briefly to inject the sample into the NMR tube. The line TTL1 outputs a TTL signal that triggers the NMR acquisition, the TTL2 signal triggers a scope which measures optical signals and the tunnel current.

## Appendix D: Bullets - Before and After

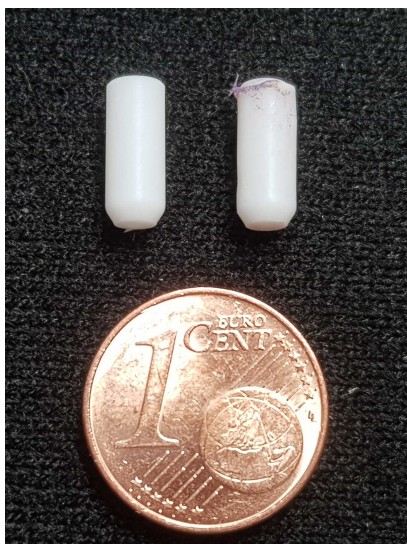

**Figure D1.** A bullet before (left) and after (right) the shot. Deformations are visible at the top and at the side. The bullets have an opening at the top and an internal volume of approximately 50 μL.

*Author contributions.* K.K. and B. M. conceived the idea, designed, built (and debugged) the hardware, performed experiments and wrote the paper. H.K. contributed to the design. M.G. performed experiments and debugged the hardware. M. J. performed experiments and co-wrote the paper.

*Competing interests.* Benno Meier, Hana Kouřilová and Karel Kouřil are co-founders and directors of HyperSpin Scientific UG. The other authors declare no competing interests.

*Acknowledgements.* When we first presented bullet-DNP at EUROMAR 2018 in Warsaw, the jubilee Geoffrey Bodenhausen pointed to our work in his lecture and guessed that we would probably be building a machine gun next. While we are rather peaceful scientists, we very much agree that the throughput of dissolution-DNP should be increased and are thankful for his early and unconditional endorsement. Now that we have embarked together with Geoffrey on the HiSCORE project, we do very much look forward to combine bullets and drugs to find new riches.

We thank Kostya Ivanov for inviting us to contribute to this special issue in the first place. Sadly, we only knew each other for a short time. We thank Fabien Ferrage and Daniel Abergel for seeing this issue through.

We thank Burkhard Luy for inviting us to join the Institute of Biological Interfaces 4 at KIT, and for his unwavering support of our activities as we were starting up.

    Finally we would like to thank Andrew M. Hall for pointing us to the existence of TgK Scientific Ltd.

    This work has been funded by the EPSRC, UK under grant agreement no. EP/R031959/1, and by the "Impuls und Vernetzungsfond of the Helmholtz Association" under grant VH-NG-1432. This project has received funding from the European Research Council (ERC) under the
European Union's Horizon 2020 research and innovation programme (grant agreement No 951459).

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
