# Peer review of "A cryogen-free, semi-automated apparatus for bullet-dynamic nuclear polarization with improved resolution"

_Magnetic Resonance, 2021_

## Author Response (AR1)

We would like to thank the referees for their comments.

**Referee #1**

Response to specific comments.

1) There are indeed three lines connected to the solvent reservoir but only two are currently used in the experiments. The figure shows only the lines which were used (for clarity). We have added a corresponding sentence to the paragraph in question.

2) We now mention the material specifically in section 2.5. We also compare the chosen material to other materials.

3) It is correct that the throughput of the system is limited by the polarization step. We have added corresponding information to the discussion.

**Referee #2**, response to the general comments:

1) We now discuss a set of two consecutive experiments in the appendix. For the discussion of throughput see the above response to comment 3 by Referee #1.

2) At this stage the reproducibility of the resolution is not very high. We now discuss the reproducibility in the manuscript in more detail.

3) We also provide a more detailed description of the cleaning procedure.

Response to the specific comments:

1)  While there are systems where D-DNP is clearly the only viable option, in other cases several days of signal averaging might be preferable to running a DNP experiment: the risk of failure is low, the required hardware and expertise are already there, the staff time needed is negligible. In addition long T1s can often be shortened using a relaxation agent. We prefer not to change the manuscript.

2) We agree with the referee that the HyperW experiments can be implemented with low dilution. We have added a sentence to the introduction and we cite the corresponding papers.

3) We have added a citation to our previous work where we used hyperpolarized solids with a volume of only 10 uL.

4) We now cite a selection of D-DNP review articles as well.

5) We agree and we have added citations to the works of Baudin et al. and Kress et al.

6) A non-decoupled experiment has been reported in our previous Nature Communications publication. At the resolution available now, the J couplings to the protons become apparent, however at the same time they may hamper a precise determination of the available resolution.

7) The first spectrum has substantially broader lines so the peak height is correspondingly lower.

8) We now report preliminary data in the appendix in which the sample tube is evacuated and the first spectrum exhibits an acceptable linewidth.

9) We thank the referee for spotting this. The correct figure is 1.75 s, as can be seen by adding the times from "pilot" up to, but not including, "Close, NMR" in the pulse program figure. We have updated the manuscript accordingly.

10) We attribute these bumps to radiation damping. We now mention this possibility in the manuscript.

11) A detailed assignment of the peaks (there are many) has been given in our Nature Communications paper, and a reference has now been added to the caption.

Minor comments:
The 2 in "Bruker NMR spectrometer 2" was an ill-formated figure reference - this is fixed.
We will report on experiments involving other molecules in due course.